# Designing a multidimensional vulnerability index for supervising dengue cases from 2015 to 2020 in a low/middle-income country: A spatial principal component analysis

Sergio Moreno-López[1,2]*, Lucia C. Pérez-Herrera[2,3,4], Augusto Peñaranda[2,4,5]

1 PhD in Public Health, School of Medicine, Universidad El Bosque, Bogotá, Colombia, 2 School of Medicine, Universidad de Los Andes, Bogotá, Colombia, 3 Department of Global Public Health, Karolinska Institutet, Stockholm, Sweden, 4 Otolaryngology and Allergy Research Groups, UNIMEQ-ORL, Bogotá, Colombia, 5 Department of Otolaryngology, Fundación Santa Fe de Bogotá, Bogotá, Colombia

* sm.morenoluniandes@gmail.com

## Abstract

### Background

Dengue is one of the most prevalent infectious diseases worldwide, affecting around 390 million people each year. Previous studies have reported that social, climatic, and government-related conditions can increase the frequency of dengue events in some territories. This study aimed to design a multidimensional vulnerability index encompassing social, climatic, and government-related factors associated with dengue and correlate this index with dengue incidence in Colombia between 2015 and 2020.

### Methods

Observational, ecological, longitudinal study conducted from 2015 to 2020. Based on administrative data from state sources such as the Ministry of Health, the National Administrative Department of Statistics (DANE), the National Planning Department (DNP), and other sources, a principal component analysis was performed to design the multidimensional vulnerability index.

### Results

Data from 1099 municipalities over the six-year analysis period were included. The index comprised five main factors: climatic factors, basic service coverage, precipitation-related factors, municipal performance, and transparency in social development. The proposed index showed a mean vulnerability of 0.48 (median = 0.48; SD = 0.15; IQR: 0.36-0.59). Higher index values were found in the southwestern territories and the Amazon regions of Colombia, as well as some

**Data availability statement:** All relevant data are in the manuscript and its supporting information files.

**Funding:** The author(s) received no specific funding for this work.

**Competing interests:** The authors have declared that no competing interests exist.

municipalities in the Caribbean region. These territories exhibited the highest levels of poverty, regional access to services, precipitation, and temperature. Spatial analyses confirmed this concordance. The nonlinear association between the MVI and dengue incidence suggests threshold effects, in which municipalities with MVI scores above 0.8 have higher levels of dengue morbidity.

## Conclusions

The proposed index showed a suitable correlation with dengue case frequency at a regional level and could be extended to other countries for the development of dengue outbreak prevention campaigns.

## Author summary

Dengue is a mosquito-borne viral disease causing millions of infections each year, with significant health impacts in low- and middle-income countries. Vulnerability to dengue increases in settings with poor access to basic services, poverty, under-resourced public health, and limited capacity of institutions to prevent and respond to disease outbreaks. Moreover, environmental factors like high temperatures and rainfall create habitats favoring the growth of mosquitoes that transmit the disease. In this study, the authors designed a Multidimensional Vulnerability Index to measure how these combinations of climatic, social, and governance-related factors influence the risk of dengue outbreaks. Using data from over 1,000 Colombian municipalities between 2015 and 2020, the authors found that areas with higher index scores, especially in the Amazon, Pacific, and Caribbean regions, had higher numbers of reported dengue cases. As climate change raises global temperatures and expands the habitats of dengue-carrying mosquitoes, dengue may emerge in countries and regions where it was not previously a public health concern. The proposed index may serve as a valuable tool for identifying high-risk areas and supporting targeted prevention strategies. While developed in the Colombian context, the framework can be adapted to other countries facing challenges of infectious disease risk and social vulnerability.

## Introduction

The World Health Organization (WHO) estimates that 3.9 billion people in 128 countries are at risk of infection with dengue virus [1]. Furthermore, around 390 million infections (95% CI: 284–528 million) are estimated per year, of which 96 million (67–136 million) have clinical manifestations and an associated mortality rate of 2.5% [2,3]. These statistics are particularly important, in terms of public health, considering the expanding range of vectors, linked to climate change, and the socio-economic vulnerabilities of at-risk populations [4,5]. Moreover, the incidence of the disease has

increased worldwide, leading to a higher burden on healthcare systems, particularly in low- to middle-income countries (LMICs) [1,6]. Furthermore, social, economic, physical, and environmental conditions can modify the epidemiological risk and lead to a higher incidence of infectious diseases in LMICs [7].

Prior studies have documented the impact of different types of vulnerability on the epidemiology of dengue [8,9]. The concept of "vulnerability" refers to the net balance of risk effects and protective and healing factors (socially, biologically and in terms of health literacy and health care access) arising from natural or anthropogenic events [10,11]. Such vulnerability can be exacerbated by social, political, and economic changes that interact at both local and international levels [12]. It has been reported that socio-economic vulnerabilities are associated with a higher frequency of communicable infectious diseases [9,13,14]. Thus, scenarios of social inequality, unequal income distribution, high migration rates, housing deficiencies, limited access to public services and health prevention programs can be considered as conducive environments for increased dengue transmission [9,13–15]. In LMICs from Latin America, the lack of access to safe drinking water and sanitation services significantly impacts dengue transmission as well [16,17]. The use of non-potable water sources and their storage methods have increased dengue transmission in Latin American countries [16,18–21]. Moreover, institutional inefficiency and lack of transparency in basic service coverage exacerbate this situation in LMICs [22,23].

On the other hand, climate change has impacted the world's ecosystems at both pathogenic and vector levels, increasing the incidence of vector-borne infectious diseases [24–28]. As climate change continues to raise global temperatures, dengue and other vector-transmitted tropical diseases (i.e., malaria, yellow fever, Chagas, etc.) may become a concern in countries and regions where they were not previously endemic. A prior study reported an expansion of the distribution of the dengue vector (Aedes aegypti mosquito) into new territories worldwide due to increased precipitation and high temperatures [29]. These climate changes have modified the diurnal temperature range (DTR), affecting the survival and adaptability of the Aedes aegypti, leading to a reduction in mortality and increased feeding frequency [30,31]. Furthermore, changes in climatic events such as precipitation, humidity, and temperature affect vector reproduction, development, mortality, viral replication within the mosquito, and the number of reservoirs [24,31–33]. Overall, an increase in dengue frequency has been described in low-income areas, suggesting an association between socioeconomic, climatic, and dengue incidence vulnerabilities [34–37]. All these social, economic, climatic/environmental, or governmental vulnerabilities may modify the epidemiological dynamics of dengue on a global perspective [31]. The interaction between these factors and the health determinants can modify both the spatial distribution patterns of the vector and the dengue transmission rate [38–43]. Although most available evidence has quantified the individual impact of specific vulnerabilities on the frequency of dengue, to date, there are no comprehensive analyses that assess the impact of these vulnerability factors on the occurrence of the disease [15,44–47]. In this study, we characterized the vulnerability factors (sociodemographic, climatic, and governmental) associated with dengue, and their relationship with the frequency of dengue in Colombia, a Latin American LMIC, during 2015–2020. This, based on a vulnerability concept defined as the presence and configuration of risk. The aim of this study was to design a multidimensional vulnerability index that synthesizes these vulnerability factors, as well as to correlate the values of this proposed index with the reported dengue case frequency in the country during the period 2015–2020.

## Materials and methods

### Ethics Statement

This study was approved by the Ethics Committee of the Hospital Universitario Fundación Santa FE (CCEI-15374–2023).

### Study design

Analytical, ecological, observational study with repeated measures based on data collected from several national sources, including: the Bank of the Republic (central bank of Colombia), the National Ministry of health, the National Administrative Department of Statistics (DANE), the National Planning Department (DNP), and the National Health Institute (INS)

[48–50]. Monthly data from January 2015 to December 2020 were analyzed, covering a total of 1,110 municipalities in Colombia. The study population included the entire population with reported dengue cases at the municipal level during the specified period.

Colombia has a population of 51,049,498 inhabitants according to the 2018 national population census. The country is divided into 32 departments, which function as first-level territorial entities, and 1,099 municipalities, which are second-level administrative divisions within departments. Each municipality has local governance structures, including a mayor and municipal council. Public health policies, including dengue surveillance and control, are implemented at both the national (Ministry of Health) and local (municipal health secretariats) levels. Local governments collaborate with the National Health Institute (Instituto Nacional de Salud or INS) and regional health institutions to monitor diseases like dengue [49,51]. Most of Colombian territory ranges between 1,000 and 2,000 meters above sea level, with average temperatures ranging from 11 to 17°C. All this information is registered in National databases from the DNP [50]. In terms of human development, Colombia ranks in the 88th position worldwide and had a Human Development Index (HDI) of 0.756 in 2020, categorizing it as having a middle to high level of human development. However, the country has significant inequalities, exhibiting a Gini index of 0.553 of 2023 (the Gini index determines a nation's level of income inequality by measuring the income distribution or wealth distribution across its population) [52].

### Dengue case data

In Colombia, dengue surveillance is conducted through the National Public Health Surveillance System (SIVIGILA) managed by the INS and conducted using passive and active surveillance with the regional healthcare institute [49,51]. The data used in this study was extracted from this system: the authors gathered all the weekly dengue reports at a municipal/ local level based [49,51]. For the analysis, the data were aggregated into annual case counts per municipality to facilitate temporal and spatial comparisons. Dengue cases are classified by SIVIGILA as probable, confirmed, or confirmed by epidemiological link, based on clinical criteria and laboratory confirmation, following national epidemiological surveillance guidelines [49,51].

### Descriptions of the variables used to design the Multidimensional Vulnerability Index

The main variables included in the design of the vulnerability index were based on prior reports and guidelines to assess vulnerability in public health [15,44,45,53–56]. Variables related with access to basic services were included, such as: percentage of population with aqueduct coverage, sewerage coverage, education coverage, healthcare coverage subsided by the government, and access to public services. Moreover, international indexes to assess the standard of living worldwide were included: Gross Domestic Product (GDP) per capita (USD millions) which is a measure of a country's economic activity (total monetary value of all goods and services produced within a country per year), and the multidimensional poverty index that quantifies poverty through weighted deprivations across health, education, and living standards (identifies individuals as poor when their deprivation score exceeds a predefined threshold). Furthermore, the latitude, the levels of relative humidity, temperature, and precipitation were assessed in the index, considering the prior literature that suggests an association between these factors and the frequency of dengue infection rates [46,47,56,57]. This information was obtained from the DNP databases and Climate Change Knowledge Portal [50,58]. Additionally, two Colombian indexes were initially included in the analysis considering their correlation with the transparency of public health campaigns for the populations. The Municipal Transparency Index (MTI) is a civil society initiative that measures the institutional conditions of city governments that could favor acts of corruption in administrative management, preventing these events [59]. On the other hand, the Municipal Performance Measurement (MPM) measures the municipal performance, understood as the management of Territorial Entities and the achievement of development results (the improvement in the quality of life of the population), considering the initial capabilities of the municipalities [60]. To achieve this measurement, the National Planning Department designed and developed a methodology structured according to the measurement components

(Management and Results) and the composition of the initial capacity group, based on which municipal comparisons and rankings are made municipalities [60].

## Design of the multidimensional vulnerability index

Data sources were collected from state databases of the DANE [48], Climate Change Knowledge Portal [58] and DNP [50]. All variables were collected assuming that they are potential indicators of vulnerability, following the recommendations to design vulnerability indices established by the "Gesellschaft Für Internationale Zusammenarbeit" (GIZ) [61] and the Organization for Economic Co-operation and Development (OECD) [62,63]. Moreover, factors associated with the dengue phenomenon were included considering a prior literature review, and reports from the WHO [1,56,64].

All variables were standardized prior to analysis. To identify patterns of covariation among variables and construct a multidimensional vulnerability index, initially we were going to apply a Principal Component Analysis (PCA), a widely used method for synthesizing correlated indicators into composite indices in epidemiological, environmental, and disaster risk contexts [63,65–70]. Although there are other approaches for multidimensional vulnerability assessment such as independent component analysis, confirmatory factor analysis or latent class analysis, PCA allows for the synthesis of information without making any assumptions about data distribution or latent structures. PCA offers advantages such as dimensionality reduction, objectivity in weighting, and applicability without assuming latent structures or specific data distributions. Although there are other approaches to multidimensional vulnerability assessment, such as independent component analysis, confirmatory factor analysis, or latent class analysis, PCA allows information to be synthesized without making assumptions about data distribution or latent structures. This analysis has been applied in previous studies to integrates socioeconomic indicators through PCA to map vulnerability to environmental hazards [71], assess vulnerability in disaster risk assessment [65–67,71], public health infrastructure risk [68], socioeconomic status [72], and the design of epidemiological indices [65,69,70].

The process of constructing the multidimensional vulnerability index (MVI) was based on the evaluation of the relationships between variables, using Spearman's correlation analysis, the Kaiser-Meyer-Olkin (KMO) value greater than 0.5 and Bartlett's test of sphericity, both of which supported the use of this method, ensuring the suitability of the variables for dimensionality reduction [62,73,74]. These variables were chosen based on a literature review identifying climatic and non-climatic variables associated with vector-borne diseases [15,28,31,34,44,45,56]. Following the guidelines established in the literature for index building and design in public health [63], a min-max normalization process was implemented to standardize the selected variables for index construction:

$y_{i,t} = \frac{x_{i,t} - Min_{x_t}}{Max_{x_t} - Min_{x_t}}$ If there is a positive relationship with the vulnerability domain.

or

$y_{i,t} = \frac{Max_{x_t} - x_{i,t}}{Max_{x_t} - Min_{x_t}}$ If there is a negative relationship with the vulnerability domain.

Where $y_{i,t}$ is the value of the normalized variable for municipality $i$ in a period $t$, and $x_{i,t}$ is the actual value of the variable for municipality $i$ in a period $t$, $min_{xt}$ and $max_{xt}$ are the minimum and maximum values for variable x in a period $t$. Given the possible spatial autocorrelation structure, a univariate Moran's I was calculated for each variable to assess the degree of spatial autocorrelation. Given the high spatial autocorrelation observed in several variables (e.g., Moran's I > 0.5, p < 0.001) led us to apply a spatially constrained Principal Component Analysis (spPCA), since the strong spatial correlation identified by Moran's analysis, if ignored, can limit the results of PCA, as the possible spatial dependence in the analyses should not be ignored when performing PCA for geographically distributed data [75]. Therefore, a spPCA was implemented, which extends traditional PCA by integrating a spatial correlation. Unlike traditional PCA, spPCA incorporates spatial structure through Moran's eigenvector maps (MEM), which improves the detection of geographically structured patterns [75]. No varimax or other orthogonal factor rotations were performed, as this would alter the spatial structure captured by the Moran's Eigenvector Maps (MEM), reducing the interpretability of the spatial components [75,76].

Finally, the final weight of each standardized variable was calculated using the squared loadings of the variables within each selected component multiplied by the proportion of variance explained by that component. This weight was

normalized across all variables to ensure that the total contribution was 100% [62]. Mathematically, the weight for a variable $y_k$ in component $c$ is:

$$\omega_k = \frac{(\lambda_c \cdot l^2_{k,c})}{\sum_k (\lambda_c \cdot l^2_{k,c})}$$

Where $\omega_k$ is the weight of variable $y_k$, $\lambda_c$ is the variance explained by component $c$ and $l^2_{k,c}$ is the loading of variable $y_k$ in component $c$. After the calculation of the weights, the multidimensional vulnerability index was defined as follows:

$$MVI = \sum_{i=1}^{k} \omega_k \cdot y_k$$

Where $\omega_k$ = Weight variable for the domain, $k$ for the number of variables included in the analysis. Both the index values and the number of cases were plotted to represent the spatial variation of the index in Colombia during each period. Scores were standardized to range from 0 (least vulnerable) to 1 (most vulnerable) at the municipal level. Five vulnerability categories were defined based on the calculated index value: very low (0-0.20), low (0.21-0.40), moderate (0.41-0.6), high (0.61-0.8), and extremely high (0.81-1) [61,75]. The spPCA analysis was conducted using the 'ade4' and 'adespatial' R packages [77,78] for incorporate spatial structure into the component extraction through MEMs [75].

### Correlational analysis of the multidimensional vulnerability index and dengue cases

To assess the joint spatial distribution of dengue cases and the Multidimensional Vulnerability Index (MVI), a bivariate Moran's I analysis was conducted to detect possible spatial autocorrelation. This method quantifies the degree to which high (or low) values of dengue cases co-occur spatially with high (or low) values of the MVI. The statistical significance of Moran's I was evaluated, and 999 permutations were used to ensure the robustness of the results. Additionally, a bivariate thematic map was created to visualize the spatial relationship between both variables. The maps represent the number of dengue cases in shades of blue and the MVI in shades of red. The resulting color gradients (e.g., purple tones) indicate municipalities where both variables exhibit high values simultaneously. On the other hand, to analyze the relationship between MVI and the number of dengue cases reported in the country from 2015 to 2020, a Spearman correlation analysis was first conducted to assess the strength and direction of their linear association. Since the results suggested potential non-linear relationships, a spatial Generalized Additive Mixed Model (spatial GAMM) with a negative binomial distribution was fitted to evaluate the association between dengue cases and the MVI while accounting for spatial and temporal structure. The model included a smooth function of the MVI, a two-dimensional spline over geographic coordinates (longitude and latitude), and a random effect for year. The spatial GAMM estimations were conducted using the 'mgcv' R package [79], applying penalized cubic regression splines to allow for flexible smoothing of the relationship between MVI and dengue incidence over time. Moran analysis was conducted using the 'spdep' package [80]. Statistical analysis was conducted using R 4.1.1 and Stata 17 MP software. The geographic base layer used for the maps (Colombian municipal boundaries) was sourced from the publicly available shapefile provided by the Línea Base project. The shapefile used is available from: https://lineabase.com.co/shape-municipios-colombia/.

## Results

### Sociodemographic, economic, and climatic characteristics of the study population

A total of 1099 municipalities were included in the study and analyzed across six years (2015–2020), yielding an initial dataset of 6594 observations that after data cleaning led to a final dataset that included 6520 observations. Table 1 describes the characteristics of these municipalities by years. The year 2019 had the highest incidence or new dengue

**Table 1. Table of sociodemographic characteristics of the study population.**

| Variables[a] | 2015 | 2016 | 2017 | 2018 | 2019 | 2020 |
|---|---|---|---|---|---|---|
| Total number of reported cases of dengue fever | 91378 | 96592 | 24448 | 42427 | 118956 | 47997 |
| Mean of reported cases of dengue fever (SD) | 6.94 (45.68) | 7.32 (75.10) | 1.85 (12.68) | 3.22 (17.18) | 9.01 (43.13) | 3.79 (17.96) |
| Height above mean sea level | 1086.07 (900.86) | 1088.79 (902.29) | 1088.79 (902.29) | 1089.74 (902.15) | 1088.79 (902.29) | 1087.84 (916.85) |
| Population density (hab/km²) | 147.02 (678.02) | 149.02 (689.36) | 151.55 (702.85) | 155.05 (720.54) | 159.32 (743.62) | 161.03 (769.87) |
| Aqueduct coverage (%) | 60.31 [35.94-88.49] | 57.48 [34.36-82.23] | 58.60 [36.44-82.78] | 75.17 [56.38-87.64] | 60.94 [37.96-85.03] | 60.94 [38.46-84.66] |
| Sewerage coverage (%) | 36.22 [17.41-62.14] | 35.06 [16.57-60.23] | 36.53 [18.48-59.71] | 44.62 [27.31-65.06] | 38.52 [19.89-61.76] | 39.56 [20.54-60.55] |
| Education coverage (%) | 83.72 [72.21-95.00] | 83.24 [71.85-95.68] | 82.56 [70.69-95.67] | 81.68 [69.63-96.27] | 88.41 [80.23-96.06] | 87.62 [79.66-94.95] |
| Healthcare coverage subsided by the government (%) | 97.60 [96.68-98.30] | 98.64 [97.84-99.12] | 98.94 [98.15-99.33] | 98.42 [97.67-98.90] | 99.36 [98.74-99.62] | 98.86 [98.13-99.26] |
| Transparency of the project: management index (%) | 73.65 [62.22-86.44] | 73.62 [62.20-86.43] | 83.78 [64.06-95.50] | 86.43 [67.46-96.53] | 85.91 [64.65-97.07] | 85.19 [65.72-96.39] |
| Access to public services (%) | 45.74 [38.72-53.88] | 45.74 [38.71-53.85] | 46.75 [39.74-53.92] | 47.00 [39.10-55.10] | 47.22 [39.74-54.38] | 48.08 [40.47-55.40] |
| Municipal Performance Measurement (%) | 47.31 [41.41-53.42] | 47.30 [41.45-53.40] | 48.95 [43.29-54.69] | 49.80 [44.36-55.71] | 55.29 [48.89-60.64] | 50.56 [44.00-57.00] |
| Multidimensional poverty index (%) | 39.98 [33.48-47.99] | 39.98 [33.50-47.97] | 34.50 [25.89-47.15] | 33.35 [25.64-53.14] | 45.18 [35.02-56.33] | 31.61 [24.06-48.16] |
| GDP per capita (USD millions)[b] | 5955.65 [3752.64-6859.33] | 5390.72 [3672.05-6613.90] | 5802.65 [3914.63-6953.09] | 5990.51 [4037.03-7469.73] | 5795.29 [3688.53-7074.89] | 4495.30 [2707.69-5677.39] |
| Precipitation (mm) | 168.19 (108.21) | 201.09 (128.06) | 180.68 (115.82) | 172.80 (110.72) | 173.65 (117.79) | 175.87 (109.77) |
| Maximum temperature (degrees Celsius) | 27.46 (3.49) | 27.31 (3.60) | 26.86 (3.56) | 26.78 (3.50) | 27.23 (3.48) | 27.25 (3.62) |
| Minimum temperature (degrees Celsius) | 18.60 (4.05) | 18.76 (3.94) | 18.47 (3.91) | 18.36 (3.92) | 18.70 (3.94) | 18.57 (3.99) |
| Relative humidity (%) | 79.36 (6.48) | 80.68 (6.84) | 80.81 (6.35) | 80.67 (7.02) | 80.17 (6.86) | 79.48 (7.18) |

[a.] Data are presented as mean (standard deviation) or median (min-max) for quantitative variables.

[b.] Based on mean TRM for each year.

cases (118956 total cases; mean = 9.01 cases, SD = 43.13 cases) with a mean of 4 cases by territory (average per year). The median altitude above sea level was 1,010 meters (range: 1.00-3,850 meters). The median values of health, and education coverage indicators were above 50% during the analyzed periods, contrasting with the median water supply coverage, which was below this value.

Regarding the distribution of dengue at the territorial level during 2015–2020, a higher number of cases are observed in the Caribbean region (Northern coastal region of Colombia located contiguous to the Caribbean Sea, mainly rural tropical regions) of the country, as well as in the central (Capital city, urban city center located in a high-altitude region) and northeastern regions (Amazonia region, rural populations). This pattern remains consistent in the territories that report the highest number of dengue cases (Fig 1).

## Multidimensional vulnerability index

To assess the spatial correlation of the selected variables, Moran's I was calculated. The results (Table 2) showed varying degrees of spatial correlation, some variables presented high Moran's I values (minimum temperature: I = 0.91;

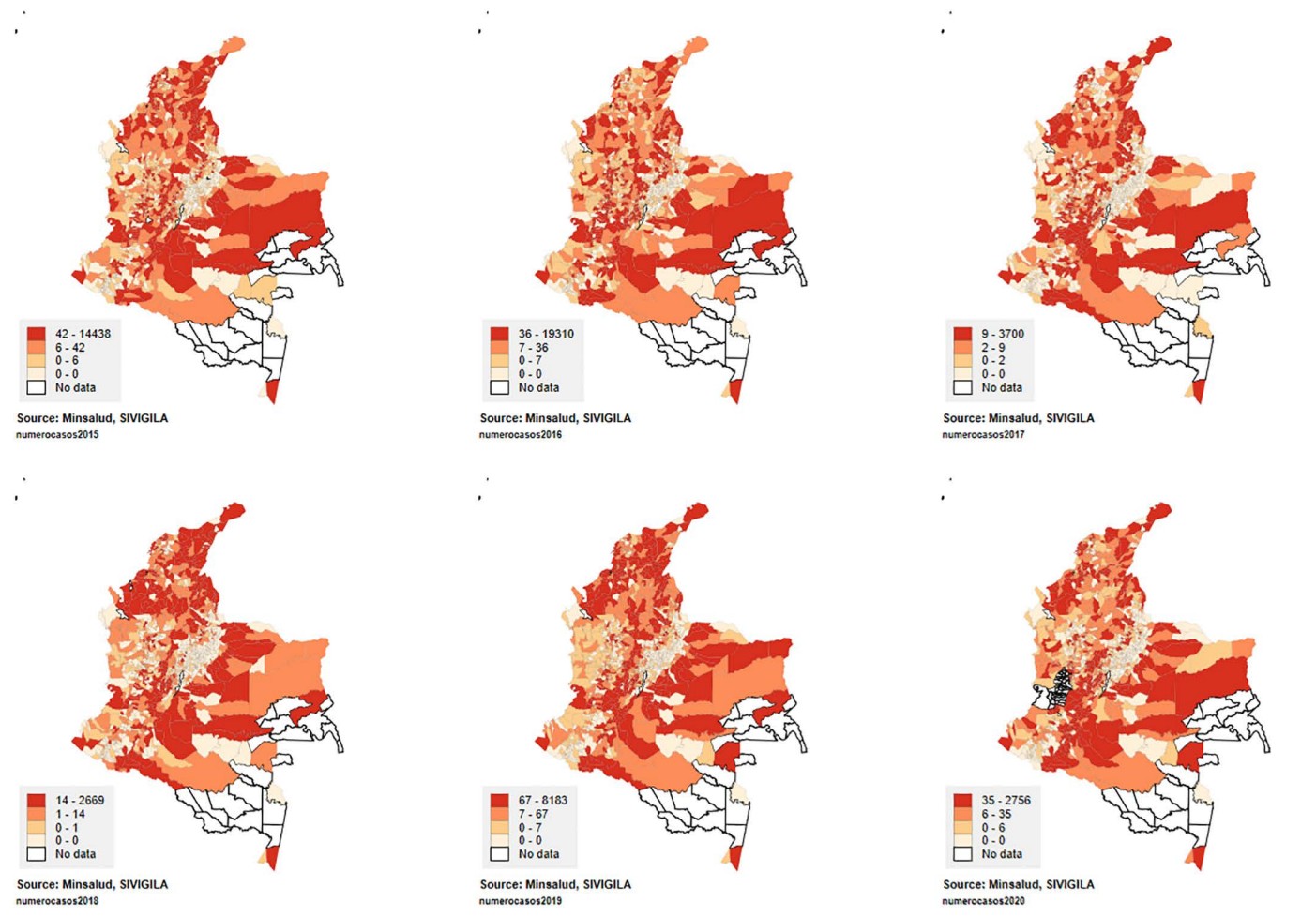

**Fig 1. Spatial distribution of reported cases of dengue fever in the study period.**

precipitation: I = 0.88) while others showed weak or no spatial correlation (access to public services: I = 0.25; education coverage: I = 0.15). These results confirm the strong spatial structure of the data, supporting the methodological decision to apply spPCA instead of traditional PCA. Of the 16 preliminary variables chosen, the indicators Transparency Index, Municipal Performance was excluded; due to high collinearity with other governmental indicators included that represent social vulnerability gradients such as Municipal performance measurement ($\rho$ = 0.67, p < 0.05), so their inclusion would have caused overrepresentation of dimensions in the index. The variables indicated in Table 2 were selected as they exhibited the highest factor loadings in the analysis, accounting for 45.55% of the total variance.

The variables after spPCA were categorized into the four domains shown in Table 2: climatic/environmental dimension, rain-related dimension, socioeconomic development, and social development dimension. The weightings give greater importance to the climatic component (weight: 45.17%) and rain-related component (weight: 27.6%), and the individual and domain-level weightings are shown in Table 3. The distribution of the variables for each domain is described in S1-S4 Figs. The proposed index showed a mean vulnerability of 0.48 (median = 0.48; SD = 0.15; IQR: 0.36-0.59). Mean IVM scores showed minimal variation across years, ranging from 0.469 (SD = 0.15) in 2016 to 0.483 (SD = 0.14) in 2018. Higher index values were found in the southwestern territories and the Amazon regions of Colombia, as well as some

**Table 2. Loadings of the Principal Components (PC) considered for vulnerability index.**

| Domain | Variable | PC 1 | PC 2 | PC 3 | PC 4 | Moran's I[a] |
|---|---|---|---|---|---|---|
| Climatic dimension | Height above sea level | 0.40 | | | | 0.802 |
| | Maximum temperature | 0.45 | | | | 0.907 |
| | Minimum temperature | 0.47 | | | | 0.910 |
| Rain-related dimension | Relative humidity | | 0.48 | | | 0.841 |
| | Precipitation | | 0.49 | | | 0.881 |
| Socioeconomic development dimension | PIB per capita | | | 0.55 | | 0.843 |
| | Population density | | | 0.22 | | 0.339 |
| | Subsidized health coverage | | | -0.03 | | 0.303 |
| | Education coverage | | | 0.07 | | 0.155 |
| | MPI | | | -0.01 | | 0.188 |
| Social development dimension | Sewerage coverage | | | | -0.27 | 0.233 |
| | Water supply coverage | | | | -0.18 | 0.172 |
| | Access to public services | | | | -0.26 | 0.252 |
| | Municipal performance measurement | | | | -0.24 | 0.231 |
| Variance explained by each PC | | 18.49 | 11.22 | 8.63 | 7.21 | -- |

[a] Moran's I calculated for each standardized variable to assess spatial autocorrelation. All values statistically significant ($p < 0.05$).

**Table 3. Domains, variables, and weights of the selected indicators for the construction of the multidimensional vulnerability index.**

| Domain | Variable | Weigh (%) | Domain weight (%) |
|---|---|---|---|
| Climatic factors | Height above sea level | 12.06 | 45.17 |
| | Maximum temperature | 16.66 | |
| | Minimum temperature | 16.45 | |
| Rain-related factors | Relative humidity | 13.2 | 27.6 |
| | Precipitation | 14.4 | |
| Socioeconomic development | PIB per capita | 14.73 | 20.42 |
| | Population density | 3.99 | |
| | Subsidized health coverage | 0.66 | |
| | Education coverage | 0.43 | |
| | MPI | 0.61 | |
| Social development | Sewerage coverage | 0.95 | 6.81 |
| | Water supply coverage | 2.24 | |
| | Access to public services | 1.98 | |
| | Municipal performance measurement | 1.64 | |

[a] Weights were obtained from the spatial PCA loadings and variance explained, standardized and rescaled following the OECD methodology for composite indicators.

municipalities in the Caribbean region. These territories exhibited the highest levels of poverty, regional access to services, precipitation, and temperature. Spatial analyses confirmed this concordance.

Regarding the territorial distribution of the index during 2015–2020, the northern and western regions of the country, as well as the region near the Amazon, exhibit the highest vulnerability values. Territories with the highest values (>0.80) are municipalities along the Pacific coast and the Amazon, while the lowest values (<0.20) were found in the Andean region, particularly in the central area of the country where main cities are located. These results remain consistent throughout the analyzed years (Fig 2).

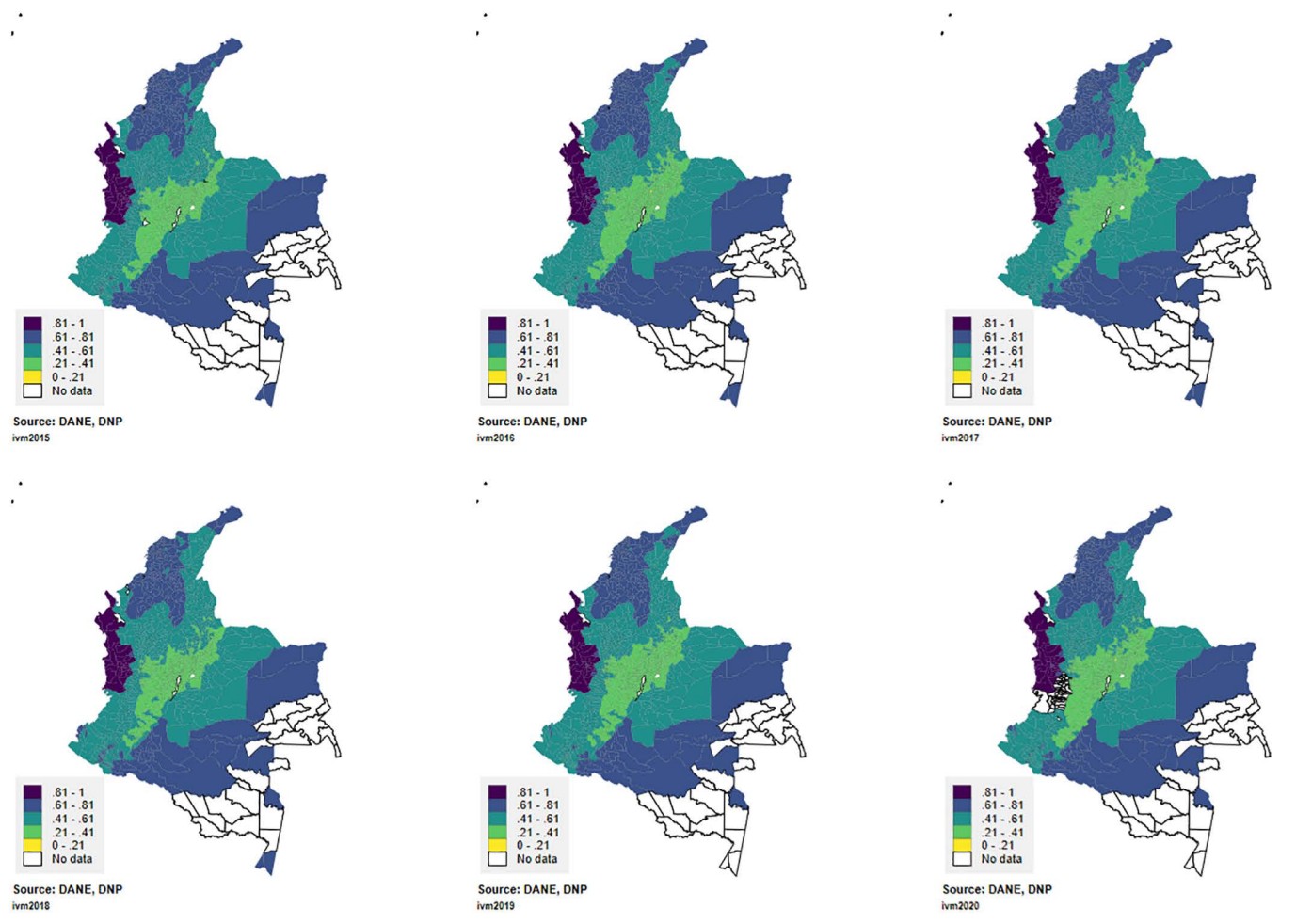

**Fig 2. Spatial distribution of Multidimensional Vulnerability Index in the study period.**

### Correlation analysis between Dengue Cases and Multidimensional Vulnerability Index

A bivariate spatial analysis was conducted to visualize the relationship between the proposed MVI and the number of reported cases in over the analyzed years (Fig 3). In the figure, the number of dengue cases is represented in shades of blue, while the proposed vulnerability index is shown in shades of red, purple tones indicate areas where both variables have high values, suggesting that territories with a higher incidence of dengue tend to align with regions classified as highly or extremely vulnerable purple tones indicate higher values in both variables, with territories reporting more dengue cases aligning with high and extremely high values of the index.

The bivariate spatial association between the Multidimensional Vulnerability Index (MVI) and dengue cases was examined through Moran's I tests on the residuals of year-specific linear regressions. All years from 2015 to 2020 showed statistically significant positive spatial autocorrelation: Moran's I ranged from 0.033 in 2015 (p = 0.027) to 0.131 in 2018 (p < 0.001), indicating that municipalities with similar levels of dengue incidence and vulnerability tend to cluster geographically. These findings support the existence of spatial structures not fully captured by the bivariate relationship, suggesting local hotspots where vulnerability and dengue incidence are concurrently elevated.

On the other hand, Spearman's rank correlation analysis between the MVI and the number of dengue cases yielded a Spearman correlation coefficient of $\rho = 0.25$ (p < 0.001), being a linear, positive and weak correlation. Additionally, the

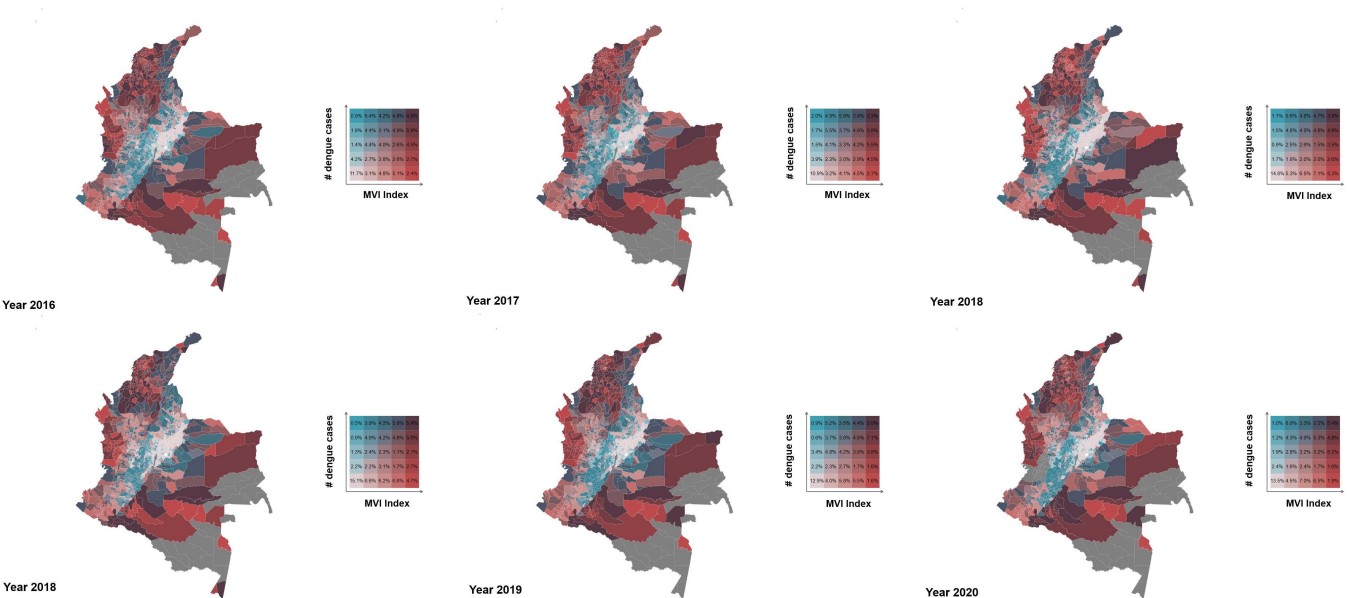

**Fig 3. Joint spatial distribution of reported cases of dengue fever and Multidimensional Vulnerability Index for the periods of analysis.**

spatial generalized additive multilevel model (spatial GAMM) showed a significant non-linear relationship between the MVI and dengue incidence (Effective Degrees of Freedom/edf = 8.7, p < 0.001), with higher case counts observed in municipalities with elevated MVI values. The spatial component was also significant (edf = 27.4, p < 0.001), confirming the presence of spatial clustering in dengue distribution. The estimated smooth function suggests that dengue cases tend to increase with higher MVI values up to a certain threshold, particularly above 0.8, after which the trend stabilizes (Fig 4).

## Discussion

This study found that municipalities with higher scores in the proposed MVI experienced significantly higher dengue incidence. The spatial Generalized Additive Model revealed a nonlinear and geographically clustered relationship, reinforcing the need to integrate social and geographic data in vector-borne disease surveillance, as priorly described in Figs 2 and 4 and the bivariate Moran's I analyses. Therefore, this index provides preliminary evidence to guide the development of outbreak prevention campaigns for this disease. The proposed index comprises four main dimensions: climatic/environmental dimension, rain-related dimension, socioeconomic development, and social development dimension. Previous studies have described that regions with specific climatic conditions and higher sociodemographic vulnerabilities require increased investment in public health to enable dengue control at the regional level [15,28,34,35,44,56,81,82].

Our findings are similar to previously published literature, underscoring the importance of the joint analysis of the conditions that may impact the epidemiology of dengue. This approach enables the development of comprehensive public health strategies that help identify the individual contributions of each dimension to the development of vulnerability conditions in populations. Despite this index included two main dimensions that are country-specific (MPM and MTI), our analysis can be extended to other countries, as these two dimensions can be replaced by local regional development measures. We highlight that all the additional measures included in the index (Climatic dimension, basic services coverage, rain-related conditions, and social development) can be universally measured. Therefore, this index can be modified and analyzed in different countries.

The proposed index showed the highest vulnerability values in the southwestern territories (Amazon and Orinoquia regions) and the Caribbean region of Colombia. According to previous literature reports, these territories exhibit the

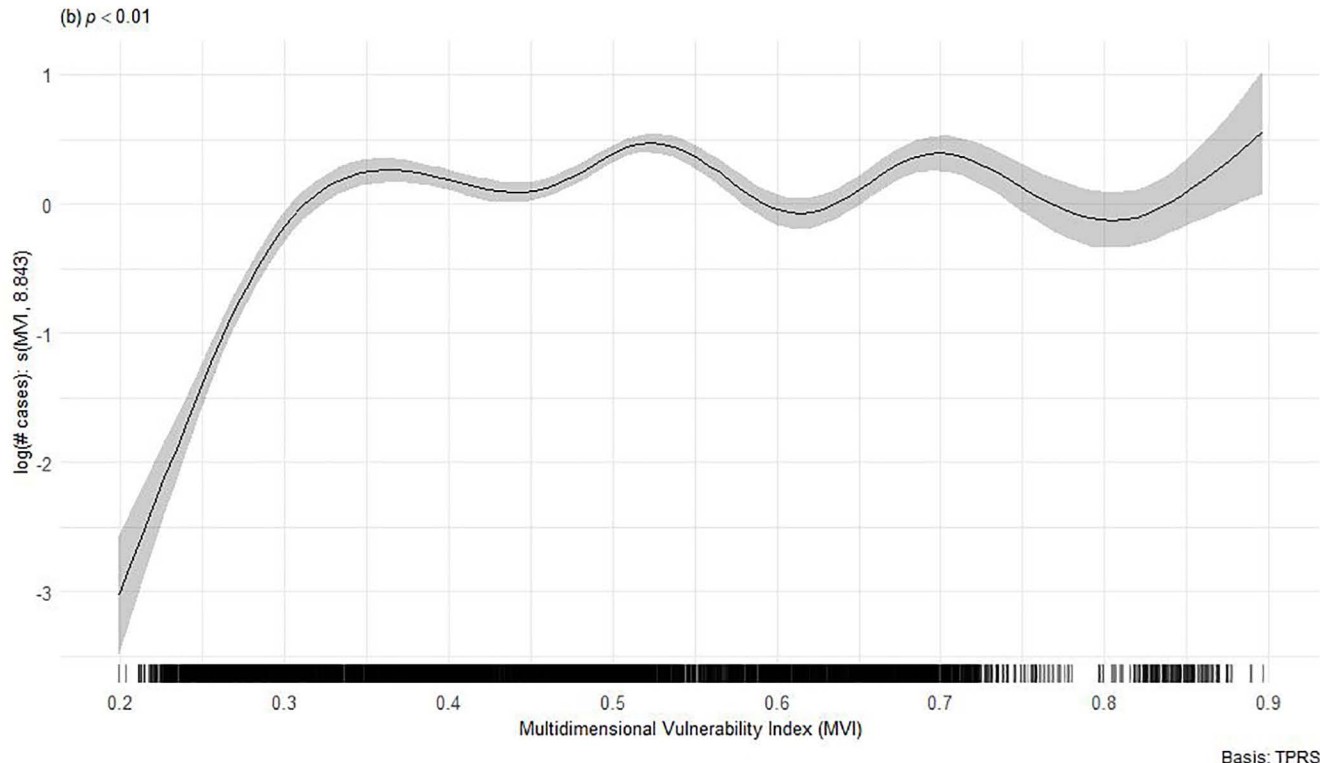

**Fig 4. Generalized Additive Mixed Models estimated for the Multidimensional Vulnerability Index and dengue cases in Colombia based on data from 2015−2020.**

highest levels of poverty, regional access to services, and increased levels of precipitation and temperature [44,83]. Spatial analyses at the municipal level further confirmed this concordance. This supports the usefulness of the index in terms of public health and allows for standardizing analyses in the case of dengue. Furthermore, we were able to characterize sociodemographic, climatic, and governmental factors and their relationship with dengue epidemiological dynamics over a 6-year period. The widespread prevalence of dengue worldwide in the last decade has led to increasing concern for identifying and controlling factors contributing to its spread [1,45,81,84]. In this scenario, our approach emerged as a proposal to understand and quantify the complex interactions underlying the dynamics of dengue. The construction of the MVI was supported by adequate psychometric criteria (KMO, Bartlett's test, and explained variance), ensuring its internal consistency. The loadings reflect theoretical constructs of vulnerability as described by Cutter et al. and Adger, and the use of domain grouping further enhances interpretability [71,85].

The use of a spatial GAMM model allowed us to validate the relationship between the MVI and the number of dengue cases, taking into account spatial autocorrelation and temporal heterogeneity. The significant spatial term shows the importance of considering geographic clustering in vulnerability and disease burden analyses. The nonlinear association between the MVI and dengue incidence suggests threshold effects, in which municipalities with MVI scores above 0.8 experience a disproportionately higher disease burden, consistent with similar findings in spatial epidemiology. These findings suggest that beyond a certain level of structural vulnerability, the ability to prevent arboviral outbreaks is significantly reduced. Similar threshold-like patterns have been observed in spatial epidemiology studies [86,87].

The MVI can be used as a tool to guide decision-making in public health planning by identifying territories at higher risk of dengue outbreaks. The spatial patterns observed highlight persistent regional vulnerabilities, particularly in the Pacific,

Amazonian, and northern regions of the country. The findings of this study highlight the need for a multisectoral approach to dengue prevention, integrating socioeconomic and environmental determinants into surveillance and response strategies. Actions such as the incorporation of the vulnerability index into dengue surveillance systems would allow its use as a predictive tool to identify high-risk areas and prioritize intervention efforts, based on theoretical matrix for the identification of the most vulnerable municipalities (Table 4).

Additionally, the MVI could also be useful for territorial epidemiologic control, allowing the identification of the risk level categories, as well as to track vulnerability trends over time, establishing real-time vector control and mitigation strategies [28,88–92]. Likewise, implementing this index, also includes the impact of climate related variables on the epidemiological trends of tropical diseases, which in the context of climate change has important implications for public health [35,56,93–95]. We highlight the need to implement policies aimed at urban planning, infrastructure improvement and equitable distribution of resources. These should be coordinated between the health, environment and urban development sectors to reduce vulnerability gradients and improve community resilience [1,3,96]. By incorporating spatial risk assessment tools like the MVI into national dengue control programs, public health authorities can move towards a proactive rather than reactive approach, enhancing the efficiency and sustainability of prevention efforts.

We highlight that the index was built considering previous methodologies for vulnerability assessment, and incorporates specific epidemiological, socioeconomic, and environmental factors related to dengue risk in the country. The index showed consistency with prior studies assessing the relationship between the Multidimensional Poverty Index and the frequency of vector-borne diseases [9,44,64,97,98]. In addition, our index also provides a more comprehensive assessment that integrates climate variability and public health infrastructure. Spatial validation using Moran's I and the nonlinear association with dengue incidence further supports its applicability. While the index is a valuable tool for identifying vulnerable regions, it should be used in conjunction with real-time surveillance systems to optimize dengue prevention strategies.

Recent studies emphasize the importance of studying dengue in "hyper-endemic" LMIC like Colombia; therefore, this study was conducted in an ideal scenario for addressing this issue [82,99–101]. However, there are significant challenges related to underreporting bias in government databases (mainly in terms of access to basic services) that have been described in the literature and can be as high as 5% [1,99,102,103]. Additionally, the municipal-level disaggregation of healthcare system affiliation variables can be considered a limitation in this analysis due to difficulties in obtaining more precise data on the existing gaps between the contributory and subsidized regimes. Despite its utility, the MVI is sensitive to indicator selection, and still lacks standardization in terms of methodology and the weighting scheme derived from spPCA. Temporal mismatch among indicators and lack of dynamic updating may affect the accuracy of the index in rapidly changing contexts. Nonetheless, we highlight that the evaluation of inequalities at the municipal level provides valuable information for guiding local public health decisions, informing about the territorial disparities that persist in our country [103]. Overall, the combination of climatic factors, service coverage, municipal performance, and transparency of social development presents an integral scope that reflects the complexity of the challenges Colombia faces in the fight against

**Table 4. Identification matrix of vulnerable territories.**

| Dengue Incidence | Low Vulnerability (MVI < 0.40) | Moderate Vulnerability (0.41 ≤ MVI < 0.60) | High Vulnerability (MVI ≥ 0.60) |
|---|---|---|---|
| Low Cases (<Percentile 25) | Low-priority areas | Potential risk zones (monitoring required) | Vulnerable but currently low-risk areas |
| Moderate Cases (Percentile 25- Percentile 50) | Areas with stable conditions | Areas requiring epidemiological surveillance | Areas requiring proactive interventions |
| High Cases (≥Percentile 75) | Epidemiological alert zones | High-risk areas requiring intervention | High-priority intervention areas |

dengue [99]. Our findings suggest that disparities in access to essential services and climatic variability may be linked to the prevalence of dengue. This scenario has been described in prior literature [24,27,104–106]

Furthermore, we highlight the spatial autocorrelation between the geographical distribution of the vulnerability index and the incidence of dengue in different regions of the country. Areas with higher social, economic, and climatic vulnerability showed a significant association with a higher frequency of dengue cases, according to spatial analyses at the municipal level. This association between vulnerability and dengue underscores the importance of specific approaches to prevent and control the spread of the disease in different areas of the country, as reported in previous studies [99]. These findings provide evidence of the relationship between multidimensional vulnerability and the frequency of dengue in Colombia. This study highlights the need for public health approaches that address both climatic and socioeconomic aspects to mitigate the spread of dengue [1]. Understanding these complex dynamics can guide effective prevention and control strategies, allowing for a more precise allocation of resources and efforts in the most affected regions. The proposed vulnerability index can be a valuable tool for prioritizing interventions and guiding dengue prevention-focused public health policies in the country. Our findings show that vulnerability is not an independent factor in outbreaks and epidemics, but rather an active determinant that influences the emergence and spread of diseases. The integration of composite indicators that incorporate spatial information into early warning systems could improve the targeting of interventions in settings characterized by socioeconomic and environmental inequalities and, in general, conditions of vulnerability.

## Conclusion

The multidimensional vulnerability index proposed in this study comprised five main factors: climatic factors, basic service coverage, precipitation-related factors, municipal performance, and transparency of social development. The index showed a strong correlation with the frequency of dengue cases at the regional level. Territories with specific climatic conditions and higher sociodemographic vulnerabilities require increased attention in terms of public health to enable dengue control at the territorial level. The proposed vulnerability index can be a valuable tool for prioritizing interventions and guiding public health policies focused on dengue prevention in the country.

## Supporting information

**S1 Fig. Correlation analysis.**
(TIF)

**S2 Fig. Distribution of the variables first domain.**
(TIF)

**S3 Fig. Distribution of the variables second domain.**
(TIF)

**S4 Fig. Distribution of the variables third domain.**
(TIF)

## Author contributions

**Conceptualization:** Sergio Moreno-López.

**Data curation:** Sergio Moreno-López.

**Formal analysis:** Sergio Moreno-López, Lucia C. Pérez-Herrera, Augusto Peñaranda.

**Investigation:** Sergio Moreno-López, Lucia C. Pérez-Herrera.

**Methodology:** Sergio Moreno-López, Lucia C. Pérez-Herrera, Augusto Peñaranda.

**Project administration:** Sergio Moreno-López.

**Resources:** Sergio Moreno-López, Augusto Peñaranda.

**Software:** Sergio Moreno-López.

**Supervision:** Sergio Moreno-López, Lucia C. Pérez-Herrera, Augusto Peñaranda.

**Validation:** Sergio Moreno-López, Lucia C. Pérez-Herrera.

**Visualization:** Sergio Moreno-López, Lucia C. Pérez-Herrera.

**Writing – original draft:** Sergio Moreno-López, Lucia C. Pérez-Herrera, Augusto Peñaranda.

**Writing – review & editing:** Sergio Moreno-López, Lucia C. Pérez-Herrera, Augusto Peñaranda.

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
