## [Decision Letter · Decision Letter 0]

3 Sep 2024

Dear Dr. Moreno-López,

Thank you very much for submitting your manuscript "Designing a multidimensional vulnerability index for supervising dengue cases from 2015 to 2020 in a low/middle-income country: A Principal Component Analysis" for consideration at PLOS Neglected Tropical Diseases. As with all papers reviewed by the journal, your manuscript was reviewed by members of the editorial board and by several independent reviewers. In light of the reviews (below this email), we would like to invite the resubmission of a significantly-revised version that takes into account the reviewers' comments.

In addition to the concerns of the reviewers, pay particular attention to the data presentation quality especially of figures.

We cannot make any decision about publication until we have seen the revised manuscript and your response to the reviewers' comments. Your revised manuscript is also likely to be sent to reviewers for further evaluation.

Sincerely,

Clarence Mang'era, PhD

Guest Editor

Paul Mireji

Section Editor

In addition to the concerns of the reviewers, pay particular attention to the data presentation quality especially of figures.

Reviewer's Responses to Questions

**Key Review Criteria Required for Acceptance?**

**Methods**

-Are the objectives of the study clearly articulated with a clear testable hypothesis stated?

-Is the study design appropriate to address the stated objectives?

-Is the population clearly described and appropriate for the hypothesis being tested?

-Is the sample size sufficient to ensure adequate power to address the hypothesis being tested?

-Were correct statistical analysis used to support conclusions?

-Are there concerns about ethical or regulatory requirements being met?

Reviewer #1: The objective of the paper is clearly stated and the introduction poses this well. Additional literature on existing multidimensional approaches is however missing.

The study design in terms of chosen data is appropriate however the technique requires a major revision. 1) The method is poorly written up with notation not defined, equations formatted poorly and the method not justified. In addition, both PCA and factor analysis are referred to - I wonder which was used. 2) The data is inherently spatial and exhibits spatial autocorrelation as can be seen in the graphics provided, thus a spatial PCA needs to be conducted, not a tradition al PCA.

Reviewer #2: 1. Are the objectives of the study clearly articulated with a clear, testable hypothesis stated?

The objectives of the study are clearly defined, focusing on the development of a multidimensional vulnerability index for supervising dengue cases from 2015 to 2020. However, the correlation between the vulnerability index and dengue cases should be explicitly stated to make the hypothesis more testable.

2. Is the study design appropriate to address the stated objectives?

The study design appears appropriate for the stated objectives. However, it is essential to include a section on the dengue surveillance system in the country, detailing how dengue data is collected and the frequency of data collection by various sectors. Additionally, clarifying whether the data was aggregated into annual summaries is necessary.

3. Is the population clearly described and appropriate for the hypothesis being tested?

The population is described in terms of territories and municipalities, but there is some ambiguity in the definitions and administrative hierarchy. A clear explanation of these terms within the country’s administrative structure is needed to ensure the population is appropriate for the hypothesis being tested.

4. Is the sample size sufficient to ensure adequate power to address the hypothesis being tested?

The sample size appears sufficient, with data spanning multiple years (2015–2020) across various municipalities and territories. However, further clarification on how the sample size relates to the administrative units and the frequency of data collection would strengthen this assessment.

5. Were correct statistical analyses used to support conclusions?

Principal Component Analysis (PCA) is an appropriate statistical method for developing a multidimensional index. However, the use of epidemiological terminologies should be consistent, and it is important to clarify whether the reported cases are annual averages, incidence rates, or monthly averages. Additionally, the calculation of "Joint spatial distribution" mentioned in Figure 3 should be explained in the methods section.

6. Are there concerns about ethical or regulatory requirements being met?

Yes.

Please include a map of the country here to highlighting dengue incidence by municipalities/territories.

**Results**

-Does the analysis presented match the analysis plan?

-Are the results clearly and completely presented?

-Are the figures (Tables, Images) of sufficient quality for clarity?

Reviewer #1: The results aren't possible to evaluate as the resolution of the graphics is to poor. This will need to be redone before it can be assessed.

I do also miss a validation of the index - to just plot the last visualisation does not validate the proposed index.

In addition, there is no discussion on the variables removed from the PCA - how does one validate the variables chosen are correct and there isn't something else that needs to be considered?

Reviewer #2: 1. Does the analysis presented match the analysis plan?

Yes

2. Are the results clearly and completely presented?

Needs improvement.

Lines 188 to 189: The statement "The year 2019 had the highest number of reported dengue cases (119,008 cases) with a mean of 4 cases by territory" is unclear. It appears that the country has more territories than municipalities (n=1100), with an average of 108 cases per municipality. Is this figure referring to the monthly average per municipality? Please clarify and revise as needed. Additionally, use appropriate epidemiological terminology (e.g., incidence, rate) and specify whether the data refers to annual average dengue cases or incidence when describing the dengue data.

In Table 1, please include the total number of dengue cases for each year and specify whether the averages provided are based on territorial or municipal data.

3. Are the figures (Tables, Images) of sufficient quality for clarity?

No.

Line 199: The Figure 1 is not clear. Please submit a clear image for review.

Line 231: The Figure 2 is not clear. Please submit a clear image for review.

Line 231: The Figure 3 is not clear. Please submit a clear image for review.

What is meant by the “Joint spatial distribution” mentioned in Figure 3? Please explain how this was calculated in the methods section. Unfortunately, the figures are unclear, making it difficult to provide further comments on this.

**Conclusions**

-Are the conclusions supported by the data presented?

-Are the limitations of analysis clearly described?

-Do the authors discuss how these data can be helpful to advance our understanding of the topic under study?

-Is public health relevance addressed?

Reviewer #1: The discussion and conclusion make a number of statements that are not shown with evidence in the results.

Reviewer #2: 1. Are the conclusions supported by the data presented?

Needs improvement. Please see my comments in methods and results sections.

2. Are the limitations of analysis clearly described?

Yes

3. Do the authors discuss how these data can be helpful to advance our understanding of the topic under study?

Needs improvement. Please see my comments below.

4. Is public health relevance addressed?

Please see my comments below.

Line 236: What is meant by “frequency of dengue cases”? Please use consistent epidemiological terminology throughout the manuscript.

Lines 277 to 278: The authors state, "Furthermore, we highlight the correlation between the geographical distribution of the vulnerability index and the incidence of dengue in different regions of the country." This is an important observation that should be supported by the methods and results. Please revise these sections to indicate how this correlation was determined.

Additionally, please provide a matrix to identify high-priority municipalities or territories, indicating the recommended operational responses.

Please include a section discussing how the vulnerability index can be integrated into dengue prevention and control strategies. Given that this composite index requires intersectoral action, it is crucial to explain how these insights can be translated into actionable steps. Additionally, address how the limitations identified in the data can be mitigated through coordinated efforts across sectors.

**Editorial and Data Presentation Modifications?**

Reviewer #1: 1) Throughout: principal component analysis does not get capital letters.

2) ln 65: These statistics are...

3) ln 121: Provide some insight to the reader on the Gini index value.

4) ln 131-132: citations missing

5) ln 139: no capitals

6) lns 141-143: there are many terms here the reader will not understand/know of. Please provide context.

7) ln 147: links needed - add as footnotes

8) ln 149: Some insight from the recommendations should be added to the paper.

9) ln 154: citation missing

10) ln 154: What does a cut off point of > 0.5 mean?

11) lns 176-180: should be in the next section

12) ln 189: How is medium altitude measured?

13) Table 1: why mean for some and median for others?

14) ln 195: Not all readers will know where these areas are?

15) ln 225: no analysis was done, only a visualisation - correct this wording. Add in the actual analysis though to validate the index.

16) ln 236: Where is this significant correlation seen?

17) ln 245-246: language should be fixed

18) ln 248: remove capitals

19) ln 254: where is this spatial analysis that is referred to here?

20) ln 264-265: it is unclear here what you want to say.

21) ln 275-276: there is not evidence provided for this statement.

22) ln 277: this correlation needs to be the spatial autocorrelation in the revision

23) ln 279-280: the results do not show this conclusively.

24) All the references need a lot of formatting and corrections. Links rather as footnotes. Many are not in English - please correct this.

Reviewer #2: (No Response)

**Summary and General Comments**

Reviewer #1: While interesting, the analysis is flawed and not well explained nor validated. A revision is required.

Reviewer #2: The manuscript presents a valuable approach by developing a multidimensional vulnerability index to guide dengue prevention and control strategies.

Please include the name of the country in the title. The term “a low/middle-income country” is not specific.

Abstract

It is not clearly stated how the vulnerability index correlates with dengue cases. Please include a statement that explains the correlation between the index and dengue incidence across the country.

PLOS authors have the option to publish the peer review history of their article (what does this mean? ). If published, this will include your full peer review and any attached files.

**Do you want your identity to be public for this peer review?** For information about this choice, including consent withdrawal, please see our Privacy Policy .

Reviewer #1: No

Reviewer #2: No
---

## [Decision Letter · Decision Letter 1]

31 Jul 2025

Designing a multidimensional vulnerability index for supervising dengue cases from 2015 to 2020 in a low/middle-income country: A principal component analysis

Dear Dr. Moreno-López,

Thank you for submitting your manuscript to PLOS Neglected Tropical Diseases. After careful consideration, we feel that it has merit but does not fully meet PLOS Neglected Tropical Diseases's publication criteria as it currently stands. Therefore, we invite you to submit a revised version of the manuscript that addresses the points raised during the review process.

Please submit your revised manuscript within 60 days Aug 30 2025 11:59PM. If you will need more time than this to complete your revisions, please reply to this message or contact the journal office at plosntds@plos.org. Please include the following items when submitting your revised manuscript:

We look forward to receiving your revised manuscript.

Kind regards,

Clarence Mang'era, PhD

Academic Editor

Paul Mireji

Section Editor

Shaden Kamhawi

co-Editor-in-Chief

Paul Brindley

co-Editor-in-Chief

**Additional Editor Comments:**

Please address the reviewers concerns with clarity and finality.

**Journal Requirements:**

**- ** Please provide an Author Summary. This should appear in your manuscript between the Abstract (if applicable) and the Introduction, and should be 150-200 words long. The aim should be to make your findings accessible to a wide audience that includes both scientists and non-scientists. Sample summaries can be found on our website under Submission Guidelines:

**Reviewers' Comments:**

Reviewer's Responses to Questions

**Key Review Criteria Required for Acceptance?**

**Methods**

-Are the objectives of the study clearly articulated with a clear testable hypothesis stated?

-Is the study design appropriate to address the stated objectives?

-Is the population clearly described and appropriate for the hypothesis being tested?

-Is the sample size sufficient to ensure adequate power to address the hypothesis being tested?

-Were correct statistical analysis used to support conclusions?

-Are there concerns about ethical or regulatory requirements being met?

Reviewer #1: Poorly provided - no actual index provided. How would it be replicated?

Reviewer #2: Authors have provided a reasonable responses to the concerns raised.

**Summary and General Comments**

Reviewer #1: The authors have provided a revised paper but I find it poorly done:

1) The response to the reviewers should be properly responded to - not just stated done or edited. Motivate what was done and why.

2) Spatial PCA is essential - the Moran's I values indicate some very high spatial autocorrelations and the authors have conveniently left out p-values here. The response to this is thus poor.

3) The authors state they have added literature on other MVI's - but there are not edits in the literature review. Later on there are some citations but that is all focussed on PCA not the design of MVIs.

4) Maths equations are poorly formatted.

5) ln 231: citation missing

6) Factor vs. principal component is still not corrected.

7) No where in the methodology is an actual index defined. The GAM is a model - then talk about a model, not an index. But the authors don't do a spatial model for inherently spatial data?? Also not fitted model discussion or information?

8) references are still poorly formatted

Overall, the authors have not put in effort to correct this paper, either ignoring review comments or working around them incorrectly.

Reviewer #2: Regarding the discussion on the utility of vulnerability index, I understand the author's caution about causality. However, from a public health application perspectives, vulnerability indices are designed to be actionable, even without establishing direct causality. The revised manuscript provides a reasonable account on it's utility. The inclusion of Table 4 is a particularly strong demonstration of the index's practical application.

PLOS authors have the option to publish the peer review history of their article (what does this mean? ). If published, this will include your full peer review and any attached files.

**Do you want your identity to be public for this peer review?** For information about this choice, including consent withdrawal, please see our Privacy Policy .

Reviewer #1: No

Reviewer #2: No

**Results**

-Does the analysis presented match the analysis plan?

-Are the results clearly and completely presented?

-Are the figures (Tables, Images) of sufficient quality for clarity?

Reviewer #1: Results are incorrect and/or not reported correctly.

**Conclusions**

-Are the conclusions supported by the data presented?

-Are the limitations of analysis clearly described?

-Do the authors discuss how these data can be helpful to advance our understanding of the topic under study?

-Is public health relevance addressed?

Reviewer #1: No valid

**Figure resubmission:**

**Reproducibility:**



---

## [Editor Report · Decision Letter 2]

10 Sep 2025

Dear Dr. Moreno-López,

We are pleased to inform you that your manuscript 'Designing a multidimensional vulnerability index for supervising dengue cases from 2015 to 2020 in a low/middle-income country: A spatial principal component analysis' has been provisionally accepted for publication in PLOS Neglected Tropical Diseases.

Best regards,

Clarence Mang'era, PhD

Academic Editor

Paul Mireji

Section Editor

Shaden Kamhawi

co-Editor-in-Chief

Paul Brindley

co-Editor-in-Chief

---

## [Editor Report · Acceptance letter]

Dear Dr. Moreno-López,

We are delighted to inform you that your manuscript, "Designing a multidimensional vulnerability index for supervising dengue cases from 2015 to 2020 in a low/middle-income country: A spatial principal component analysis," has been formally accepted for publication in PLOS Neglected Tropical Diseases.

Best regards,

Shaden Kamhawi

co-Editor-in-Chief

Paul Brindley

co-Editor-in-Chief
